# Antibody Neutralization of an Influenza Virus that Uses Neuraminidase for Receptor Binding

**DOI:** 10.3390/v12060597

**Published:** 2020-05-30

**Authors:** Lauren E. Gentles, Hongquan Wan, Maryna C. Eichelberger, Jesse D. Bloom

**Affiliations:** 1Division of Basic Sciences Basic Sciences, Fred Hutchinson Cancer Research Center, 1100 Fairview Ave N, Seattle, WA 98109, USA; lgentles@uw.edu; 2Department of Microbiology, University of Washington, 1705 NE Pacific St., Seattle, WA 98195-7735, USA; 3Division of Viral Products, Center for Biologics Evaluation and Research, Food and Drug Administration, Silver Spring, MD 20993, USA; Hongquan.Wan@fda.hhs.gov; 4Division of Biological Standards and Quantity Control, Center for Biologics Evaluation and Research, Food and Drug Administration, Silver Spring, MD 20993, USA; Maryna.Eichelberger@fda.hhs.gov; 5Howard Hughes Medical Institute, Seattle, WA 98195, USA

**Keywords:** influenza virus, neuraminidase, neutralization, antibody escape, G147R, receptor-binding

## Abstract

Influenza virus infection elicits antibodies against the receptor-binding protein hemagglutinin (HA) and the receptor-cleaving protein neuraminidase (NA). Because HA is essential for viral entry, antibodies targeting HA often potently neutralize the virus in single-cycle infection assays. However, antibodies against NA are not potently neutralizing in such assays, since NA is dispensable for single-cycle infection. Here we show that a modified influenza virus that depends on NA for receptor binding is much more sensitive than a virus with receptor-binding HA to neutralization by some anti-NA antibodies. Specifically, a virus with a receptor-binding G147R N1 NA and a binding-deficient HA is completely neutralized in single-cycle infections by an antibody that binds near the NA active site. Infection is also substantially inhibited by antibodies that bind NA epitopes distant from the active site. Finally, we demonstrate that this modified virus can be used to efficiently select mutations in NA that escape antibody binding, a task that can be laborious with typical influenza viruses that are not well neutralized by anti-NA antibodies. Thus, viruses dependent on NA for receptor binding allow for sensitive in vitro detection of antibodies binding near the catalytic site of NA and enable the selection of viral escape mutants.

## 1. Introduction

Neuraminidase (NA) and hemagglutinin (HA) are the two major proteins on the surface of influenza virions, and play opposing roles during the viral life cycle. HA mediates viral attachment and entry into cells, while NA cleaves sialic-acid receptors to release newly formed virions and prevent their aggregation [1]. NA also plays an important role in vivo by helping the virus penetrate mucus barriers to reach target cells [2,3]. Because only HA is needed for viral entry, anti-NA antibodies are not strongly neutralizing in infection assays where viruses are only allowed to undergo a single cycle of growth [4,5]. However, many studies have shown that anti-NA antibodies are associated with reduced disease severity in humans [6,7,8].

Recently, several exceptions to the classic role of NA as a receptor-cleaving but not a receptor-binding protein have been uncovered. Beginning in 2003, several groups identified mutations at NA site 151 in H3N2 clinical isolates that had been passaged in cell culture [9,10,11,12,13,14,15]. It was soon discovered that the mutations D151G/N allow N2 NA to bind sialic-acid receptors, but ablate NA catalytic receptor-cleaving activity [10,13,15,16]. It was subsequently shown that because D151G/N NA lacks enzymatic activity, viruses carrying these mutations can only grow in mixed “cooperating” populations with viruses encoding NA that retains receptor-cleaving activity [17]. Importantly, the D151G/N mutations appear to only arise in cell culture, and have not been found in actual human infections [12,14,18].

Shortly after the identification of D151G/N in N2 NA, it was discovered that the G147R mutation enables N1 NA to bind to cellular receptors while maintaining its receptor-cleaving function [19]. Viruses carrying this NA mutation can grow as a clonal population unaided by wildtype virions [19]. Unlike D151G/N mutations that only arise in N2 NA in tissue culture, the G147R mutation has been identified at low frequency in several naturally occurring H1N1 and H5N1 isolates [20]. Importantly, viruses with the G147R N1 NA can grow efficiently in cell culture even if the receptor-binding activity of HA is completely ablated by engineered mutations [19,20].

Here, we utilize the G147R NA in conjunction with a binding-deficient HA to develop a sensitive neutralization assay for anti-NA antibodies. Specifically, we test the susceptibility of this NA-binding-dependent virus to neutralization by four monoclonal antibodies targeting distinct epitopes of N1 NA [5], and find that some of these antibodies neutralize the NA-binding-dependent virus much better than they neutralize a virus that can bind cells via HA. We then leveraged this discovery to select an in vitro escape mutation to one of the antibodies, demonstrating the value of these viruses for antigenic mapping. Overall, our work suggests that NA-binding-dependent viruses may be a useful tool for studying antibodies targeting NA.

## 2. Materials and Methods 

### 2.1. Viruses and Reverse Genetics Plasmids

The viruses used in this study all had internal genes derived from A/WSN/1933. The NA was derived from A/California/7/2009 (H1N1). The binding-competent HA was also derived from A/California/7/2009 (H1N1). The binding-deficient HA is referred to as “PassMut” HA in the original reference [19] describing its creation. This HA is derived from the A/Hong Kong/1968 H3N2 strain, and has the following mutations: Y98F, H183F, and L194A at receptor binding residues; seven potential N-linked glycosylation sites added at residues 45, 63, 122, 126, 133, 144, and 246; deletion of the receptor binding proximal loop spanning residues 221–228; and mutation K62E in the HA2 stalk (all mutations in H3 numbering).

All viruses packaged eGFP in place of the PB1 coding sequence in the PB1 segment to enable easy detection by fluorescence, and so were propagated in cells expressing PB1 as described previously [21]. Briefly, the PB1flank-eGFP segment retains the non-coding regions of PB1 and the last 80 terminal nucleotides which allows for efficient packaging of this segment into virions and replication to high titers when complemented by PB1 constitutively expressing cells. These viruses were generated from the following reverse genetics plasmids: for the WSN internal genes, pHW181-PB2, pHW183-PA, pHW185-NP, pHW187-M, and pHW188-NS [22]; for the PB1 segment, pHH-PB1flank-eGFP [21]; for the NA segment either pHWCA09tc-NA or pHWCA09tc-G147R; and for the HA segment either pHW-CA09tc-HA or pHWX31-HA-NGly12-Y98F-L194A-H183F-del221to228-K62EHA2. Here, “tc” indicates that these segments were originally cloned from tissue culture-derived virus.

For experiments with the S364N mutant of NA, we introduced S364N into pHWCA09tc-NA-G147R to create pHWCA09tc-G147R-S364N.

### 2.2. Virus Rescue and Titering

All viruses were rescued in PB1flank-eGFP format by reverse genetics as previously described with minor modifications [21]. Briefly, in 2 mL DMEM, supplemented with 10% heat-inactivated fetal bovine serum (FBS) and 2 mM L-glutamine, we transfected co-cultures of 2 × 10^5^ 293T-CMV-PB1 [21] and 2.5 × 10^4^ MDCK-SIAT1-CMV-PB1-TMPRSS2 cells [23] per well in six well plates with 200 ng of each of the eight plasmids encoding the influenza gene segments plus a PB1 protein expression plasmid, pHAGE2-CMV-PB1-IRES-mCherry-W, and a TMPRSS2 expression plasmid, pHAGE2-EF1aInt-TMPRSS2-IRES-mCherry [23] to augment viral growth. At 14–16 hours post-transfection, the media was changed to 1mL of Opti-MEM supplemented with 100 ug/mL CaCl_2_, 0.3% bovine serum albumin (BSA), and 0.01% FBS, here referred to as influenza growth media (IGM). At 48 h post-transfection, the supernatant was collected, clarified by centrifugation, and frozen at −80 °C. All viruses were then expanded in 10 cm dishes at a low multiplicity of infection (MOI) (<0.1) in MDCK-SIAT1-CMV-PB1-TMPRSS2 cells that allow for multiple rounds of replication since PB1 protein is provided in trans and TMPRSS2 cleaves HA into its active form. Viruses were grown for 72 h and then collected, clarified, and stored at −80 °C.

Infectious particles were titered as described [19] with minor modifications. Briefly, 5 × 10^5^ MDCK-SIAT1-CMV-PB1 cells per well in a 12-well plate were infected with serial dilutions of viral supernatant for 14–16 h. The cells were then harvested by trypsinization and fixed for 1 h in 1% paraformaldehyde in PBS at room temperature before being washed twice and resuspended in a 1% BSA solution in PBS. The cells were then analyzed by flow cytometry to determine the % GFP positive cells, and this measurement was used to calculate the concentration of infectious particles in the viral stocks.

### 2.3. Antibodies

The antibodies used in this study were originally described in the references [5,24,25].

### 2.4. NA Microneutralization Assay

Micro-neutralization assays were performed as previously described with some modifications [19,20,26,27]. Briefly in 96-well flat-bottom plates, 5-fold dilutions of each antibody were made across each plate in low autofluorescent media (Medium 199 supplemented with 0.01% FBS, 0.3% BSA, 100 ug/mL CaCl_2_, and 25 mM HEPES buffer). We then added either the NAbind/HA∆bind virus, NAbind+S364N/HA∆bind virus, NAbind/HAwt virus, or NAwt/HAwt virus using an infectious dose determined to be within the linear range of detection. This infection dose was determined by making 2-fold serial dilutions of the viruses in 96-well flat bottom plates and infecting 4 × 10^4^ MDCK-SIAT1-CMV-PB1 cells per well to find the highest viral dose producing a GFP signal at 15–17 h post-infection within the linear range of detection (i.e., the highest dose where the signal still increased linearly with the virus concentration). To measure infectivity in the presence of each antibody, the virus/antibody mixtures were incubated at 37 °C for 1 h before adding 4 × 10^4^ MDCK-SIAT1-CMV-PB1 cells per well. Viral growth was measured 15–17 h post-infection on a Tecan^®^ Infinite^®^ M1000 PRO plate reader to detect the eGFP signal from infected cells. The background signal from wells containing virus and media only was subtracted from all measurements. All neutralization curves represent the mean and standard error of the mean from three technical replicates read from a single plate.

### 2.5. Selection of Escape Mutants

Escape mutations were selected by first incubating 10^6^ infectious particles of the NAbind/HA∆bind virus with 4ug/mL of the antibody HF5 at 37 °C for 1 h. The total volume of the virus-antibody mixtures was 1 mL in IGM. We then added the mixture to a 6-well plate containing 5 × 10^3^ MDCK-SIAT1-CMV-PB1-TMPRSS2 cells per well, rocking the plate every 15 min for 1 h to promote infection. After the 1-hour incubation, we removed the remaining virus/antibody mixture and added 2 mL of fresh IGM. At 71 h post-infection, viral supernatant was collected from the selected sample, clarified by centrifugation, and stored at −80 °C. The selected viruses were then expanded twice by infecting MDCK-SIAT1-CMV-PB1-TMPRSS2 cells with the selected viral supernatant, allowing the virus to grow for 72 h. After this expansion step, the selection was repeated as above to deplete residual wildtype virus. One final expansion of the selected virus was done as above prior to sequencing.

### 2.6. Sanger Sequencing

Viral RNA was extracted from bulk selected virus supernatant using a Qaigen QIAmp® Viral RNeasy Mini Kit according to manufacturer instructions. We then used the SuperScript™ III First-Strand Synthesis System to generate cDNA using a universal influenza RT primer with U12-G4 homology - 5’ TATTGGTCTCAGGGAGCGAAAGCAGG 3’ [28]. HA and NA gene segments were then further amplified by PCR using KOD Hot Start Master Mix and the following primers: NA forward primer—5’ TATTGGTCTCAGGGAGCAAAAGCAGGAGT 3’ [28] and NA reverse primer—3’ ATATGGTCTCGTATTAGTAGAAACAAGGAGTTTTTT 3’ [28]; HA∆bind forward primer—5’ ATGAAGACCATCATTGCTTTGAGCTACATTTTC 3’ and HA∆bind reverse primer—5’ AGTAGAAACAAGGGTGTTTTTAATTACTAATACACTCA 3’ that we designed. The PCR products were then Sanger sequenced. The consensus sequencing was compared to the original NA sequence and selected mutations were identified as those arising to the majority (>50% frequency) in chromatograms.

## 3. Results

### 3.1. Anti-NA Monoclonal Antibodies Neutralize an NA-Binding-Dependent Virus but not an HA-Binding Viruses in Single-Cycle Infection Assays

We generated three influenza viruses with differing dependencies on HA and NA for viral entry. The first virus contains the unmutated HA and NA from the pandemic H1N1 vaccine strain A/California/7/2009 (Cal09). This virus has an HA that binds receptors and an NA that cleaves receptors, and will be referred to as NAwt/HAwt throughout the rest of this paper. We expected this virus to be unaffected by NA antibodies in a single-cycle infection assay (Figure 1). 

The second virus has the unmutated Cal09 HA, but the NA has the G147R mutation relative to the Cal09 sequence. As described in the Introduction, the G147R mutation enables NA to bind receptor while retaining its catalytic receptor-cleaving activity [19,20]. This virus will be referred to as NAbind/HAwt throughout the rest of this paper. We expected this virus to also be largely unaffected by NA antibodies in a single-cycle infection assay, since HA can mediate attachment to cells even when NA binding is blocked.

Finally, we generated a virus that is totally dependent on NA for receptor binding. To do this, we used a previously described engineered binding-deficient version of the HA from the H3N2 strain A/Hong Kong/1968 [19,20]. This engineered HA contains amino-acid mutations at key residues in the receptor binding pocket (Y98F, H183F, and L194A in H3 numbering [29]), deletion of residues 221-228 that form a loop near the receptor-binding pocket [30], and the addition of seven potential N-linked glycosylation sites (at residues 45, 63, 122, 126, 133, 144, and 246) that decrease HA receptor avidity [31]. The HA also contains the mutation K62E in the HA stem domain, which has previously been shown to enhance viral growth in the context of the other HA mutations [19,20]. Note that unlike the other HAs in this paper, this binding-deficient HA is H3 rather than H1—our rationale for using it is that it has been shown to completely lack receptor binding activity [19,20], whereas most characterized point mutants of HA in the receptor binding pocket (e.g., the widely used Y98F mutant) still retain a sufficient binding activity to enable HA-mediated infection in cell culture [32]. This binding-deficient HA was paired with the G147R mutant of the Cal09 NA to create a virus that we will refer to as NAbind/HA∆bind. We expect that anti-NA antibodies may neutralize this virus in single-cycle infection assays, since it depends on NA to attach to cells (Figure 1).

We tested neutralization of all three viruses using several well-characterized antibodies (HF5, CD6, 1H5, and 4E9) targeting different epitopes on NA’s membrane-distal head domain [5]. These antibodies all bind to Cal09 NA and inhibit NA catalytic activity to varying degrees. HF5 and CD6 are strain-specific antibodies that each target distinct epitopes, and strongly inhibit NA activity as measured by enzyme-linked lectin assays (ELLA) which detect NA cleavage of the large glycoprotein substrate, fetuin [5,33]. Antibodies 1H5 and 4E9 recognize a similar epitope on the lateral surface of the NA head that is conserved across the N1 subtype, but these antibodies only weakly inhibit NA activity in ELLA assays [5].

We tested neutralization by each antibody in a single-cycle infection assay using a previously described method that uses viruses packaging eGFP in the PB1 segment [19,20,26,27]. The antibodies were pre-incubated with the virus for one hour at 37 °C, and then the virus/antibody mix was used to infect cells for 15–17 hours before reading the fluorescent signal. In order to ensure only a single-cycle of viral growth, we performed the infections in the absence of trypsin, thereby precluding the proteolytic activation of newly formed HA that is necessary to enable secondary infection [34,35].

As expected, none of the antibodies substantially neutralized the NAwt/HAwt virus (Figure 2). However, several antibodies partially neutralized and one completely neutralized the NAbind/HA∆bind virus (Figure 2). The antibody that completely neutralized the NAbind/HA∆bind virus was HF5, which binds to the top of the NA near the active site [5]. Interestingly, HF5 also partially neutralized the NAbind/HAwt virus, suggesting that it uses both NA and HA to bind to cells.

The extent to which antibodies neutralized the NAbind/HA∆bind virus paralleled with their ability to inhibit NA catalytic activity in previously reported ELLA assays (Table 1) [5]. Specifically, HF5 has the strongest ELLA activity and was the most potent in our neutralization assay, CD6 has intermediate ELLA activity and partially neutralized in our assay, and 4E9 and 1H5 have little impact on NA catalytic activity in ELLA assays and do not neutralize [5]. We speculate that the correlation between neutralization of NAbind/HA∆bind and inhibitory activity in the ELLA assay reflects the fact that the G147R receptor-binding NA binds to cells using the NA active site [19]. Therefore, an antibody that blocks NA activity would also be expected to block binding.

### 3.2. Selecting Antibody-Escape Mutations Using the NA-Binding Virus

One classic approach to map the epitopes of neutralizing antibodies is to select viral mutants that escape neutralization. This approach has been widely and successfully applied to neutralizing anti-HA antibodies where it works very well because it directly selects for viral mutants that can enter cells in the presence of the antibody [36,37,38,39]. We tested whether our NAbind/HA∆bind virus could similarly be used to select antibody viral escape mutants in NA. To accomplish this, we pre-incubated a stock of the NAbind/HA∆bind virus with the HF5 antibody at an excess of the amount needed to totally neutralize this virus, and then infected the cells with the antibody/virus mix to select for mutants that escaped antibody binding. The resulting virus was then passaged to produce appreciable titers before performing a second round of selection to deplete any parental virus remaining in the population.

Upon Sanger sequencing of the selected virus population in bulk, we discovered two mutations that were present in the consensus sequence after the antibody selection. One of the selected mutations, S364N, had previously been identified as an HF5 escape mutation in cell-based ELISAs using a panel of mutants and again after the passage of Cal09 virus in passively immunized animals [5]. The second mutation was D151N—as discussed in the Introduction, this mutation confers receptor-binding properties on N2 NA [10,13,15,16]. We speculate that D151N indirectly increases antibody resistance by enhancing NA’s affinity for receptors, similar to the “avidity” mutations that sometimes arise in HA under antibody pressure [40]. Therefore, D151N might allow the virus to outcompete antibody binding by increasing NA binding avidity to cells, allowing rapid attachment and therefore reducing sensitivity to antibodies in a similar fashion to that shown for HA [40]. Since D151N ablates NA catalytic activity, it is not possible to rescue this virus as a clonal population [17], so we only tested the S364N mutation in subsequent validation. Based on prior work [17], it may be possible to generate D151N as a mixed cooperating population, but we did not attempt that here.

To validate that S364N is an HF5 escape mutation in the NAbind/HA∆bind virus, we performed neutralization assays with the parental NAbind/HA∆bind virus and a reverse-genetics generated virus with S364N introduced into this background. As expected, S364N greatly increased the resistance of the virus to HF5 neutralization (Figure 3). This observation illustrates the utility of the NAbind/HA∆bind virus for identifying escape mutants from antibodies that target regions of NA near the active site.

## 4. Discussion

We have shown that engineered influenza viruses that use NA for receptor-binding are potently neutralized by some anti-NA antibodies. This contrasts with the typical situation for viruses that bind receptors using HA, where the inhibitory activity of anti-NA antibodies only becomes strongly apparent after multicycle growth [4,41].

Notably, neutralization of NA receptor-binding-dependent viruses is strongest for antibodies that bind to the top of the NA head near the active site where receptor binding also occurs. A similar trend occurs for HA, with the most potent antibodies often directly blocking the interaction between HA and cellular sialic acid receptors [37,39,42,43]. Because the neutralization of the NA-binding-dependent virus likely depends on blocking receptor binding by residues in NA’s catalytic site, neutralization by anti-NA antibodies parallels their ability to inhibit NA catalytic activity in ELLA assays. Of interest, a recent study identified NA antibodies that are broadly protective in mice and bind near the active site of NA [44], suggesting that the types of NA antibodies that are highly active against NA receptor-binding viruses in vitro can also be protective in vivo.

We also show that NA-binding-dependent viruses can be used to easily select for anti-NA antibody escape mutations in vitro. Previous in vitro methods of selecting escape mutants to anti-NA antibodies require selection in multi-cycle growth conditions or by passaging the virus in passively immunized animals [5,45]. Therefore, we suggest that in some cases, NA receptor-binding viruses could be a useful tool to simplify the mapping of anti-NA antibody epitopes via escape-mutant selections.

## Figures and Tables

**Figure 1 viruses-12-00597-f001:**
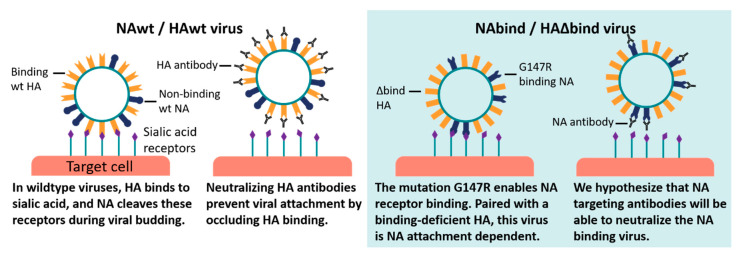
Proposed mechanism of neutralization of neuraminidase (NA)-binding-dependent viruses by anti-NA antibodies. Most influenza viruses use hemagglutinin (HA) to bind and enter cells (left panel), so antibodies targeting HA can prevent infection while NA antibodies only inhibit the release of new virions. But in our engineered NA-binding-dependent virus (right panel), antibodies against NA can inhibit infection.

**Figure 2 viruses-12-00597-f002:**
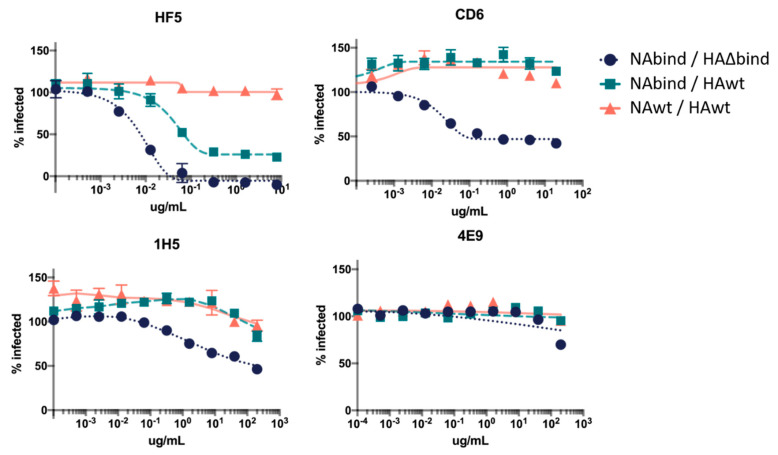
Neutralization of viruses with different dependencies on HA and NA for receptor binding by anti-NA antibodies in single-cycle infection assays. Neutralization curves show the percentage of the NAwt/HAwt, NAbind/HAwt, or NAbind/HA∆bind virus infection that is inhibited by each antibody over a range of concentrations compared to the no-antibody control infection. Points represent the mean of three technical replicates with error bars showing the standard error of the mean.

**Figure 3 viruses-12-00597-f003:**
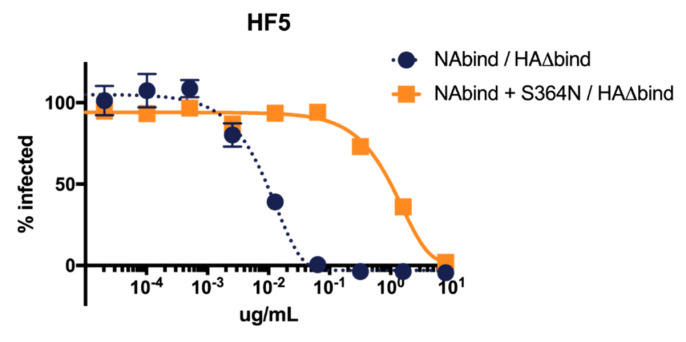
S364N increases resistance to neutralization by antibody HF5 in single-cycle infections for viruses that are dependent on NA for receptor-binding. Results are shown as the percent infection at each concentration compared to the no-antibody infection control. Points represent the mean of three technical replicates with error bars showing the standard error of the mean.

**Table 1 viruses-12-00597-t001:** NA antibody inhibition by epitope location.

Antibody	Amino-Acids Targeted	ELLA Inhibition	Neutralization Inhibition
HF5	364, 369, 397	+++	+++
CD6	95, 449, 451	++	++
1H5	273, 338, 339	+	+
4E9	273, 338, 339	+	-

(−) no activity, (+) weak activity, (++) moderate activity, (+++) strong activity. Enzyme-linked lectin assay (ELLA).

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
