# Peer review of "Antibody Neutralization of an Influenza Virus that Uses Neuraminidase for Receptor Binding"

_viruses, 2020, doi:10.3390/v12060597_

Round 1

Reviewer 1 Report

The brief report by Gentles et al. describes a new method for assessing neutralizing properties on anti-neuraminidase (NA) antibodies and antibody escape mutants using single-cycle replication reporter viruses. The design takes advantage of a previously discovered NA mutant that can confer sialic acid receptor binding properties to NA. When combined with hemagglutinin (HA) mutant lacking receptor binding activity using reverse genetic system, a reporter virus is generated that relies on NA exclusively for virus attachment to cells. Monoclonal antibody HF5 that binds at the catalytic site of NA strongly neutralized the reporter virus. When virus escape mutants were selected, two single amino acid substitutions were identified.

The study presented in the report is original and scientifically sound. There are few modifications to the text that in the opinion of this reviewer would improve the manuscript:

Lines 77-81: because of the method development nature of the report, a more detailed description of reporter virus generation and plasmid descriptions (e.g. meaning of "tc" abbreviations) would be helpful instead of just references.

Lines 94-95: please provide details of the methodology, it is unclear how a single-cycle virus stock was "expanded in 10-cm dishes".

Line 180: please describe ELLA assay principle and give a full name of the method before using abbreviation for the first time.

Lines 225-226: authors speculate that the D151N substitution "indirectly increases antibody resistance by enhancing NA’s affinity for receptor". This point is unclear and a better description is needed. Was further analysis done on S364N mutation only because it does not destroy neuraminidase activity of NA and thus makes it easier to generate? If this is the case, a description of the strategies that can be used to overcome this defect is needed.

Author Response

Responses to Reviewer 1 are written in italic:

The brief report by Gentles et al. describes a new method for assessing neutralizing properties on anti-neuraminidase (NA) antibodies and antibody escape mutants using single-cycle replication reporter viruses. The design takes advantage of a previously discovered NA mutant that can confer sialic acid receptor binding properties to NA. When combined with hemagglutinin (HA) mutant lacking receptor binding activity using reverse genetic system, a reporter virus is generated that relies on NA exclusively for virus attachment to cells. Monoclonal antibody HF5 that binds at the catalytic site of NA strongly neutralized the reporter virus. When virus escape mutants were selected, two single amino acid substitutions were identified.

The study presented in the report is original and scientifically sound.

There are few modifications to the text that in the opinion of this reviewer would improve the manuscript:

Lines 77-81: because of the method development nature of the report, a more detailed description of reporter virus generation and plasmid descriptions (e.g. meaning of "tc" abbreviations) would be helpful instead of just references.

Added a better description of how the reporter virus plasmid is expressed and propagated as well as a sentence describing the meaning of “tc” abbreviation to lines 81-82 (it stands for tissue-culture derived virus, which is where the genes were originally cloned).

Lines 94-95: please provide details of the methodology, it is unclear how a single-cycle virus stock was "expanded in 10-cm dishes".

Added clarifying statement to line 96. Essentially, the cells provide PB1 protein to complement the lack of virally encoded PB1, and the cells also provide TMPRSS2 to cleave HA into its active form during propagation (but TMPRSS2 is not present in the cells subsequently used for the single-cycle neutralization assays).

Line 180: please describe ELLA assay principle and give a full name of the method before using abbreviation for the first time.

Added full name and description of the ELLA assay to lines 188-189 along with reference to assay protocol.

Lines 225-226: authors speculate that the D151N substitution "indirectly increases antibody resistance by enhancing NA’s affinity for receptor". This point is unclear and a better description is needed. Was further analysis done on S364N mutation only because it does not destroy neuraminidase activity of NA and thus makes it easier to generate? If this is the case, a description of the strategies that can be used to overcome this defect is needed.

Added several sentences better describing the evidence that D151N enhances NA receptor binding and explained how that can enhance antibody resistance. We also explained how D151N can only be rescued as a cooperating virus population, and why that fact provided our rationale for focusing on S364N.

Reviewer 2 Report

As compared to hemagglutinin, neuraminidase and antibodies recognizing neuraminidase (NA) have received decidedly less attention. Yet, it is increasingly clear that a better understanding of NA-based immunity has important implications for the development of novel therapeutics. Innovative tools and techniques to study anti-NA antibodies are needed to help fill our gaps in knowledge. In the study described here, Gentles and colleagues describe a modified influenza A virus that is dependent on NA for entry. The experiments that follow are straightforward. Using a single-cycle infection assays, the authors demonstrate that a panel of anti-NA antibodies can neutralize this HA-binding deficient virus. The authors then go on to demonstrate that this modified IAV can be used to select antibody escape mutant for NA. The latter use offers perhaps the greatest utility going forward, given that enzyme-linked lectin assays (ELLA) can be used to identify antibodies that inhibit NA enzymatic activity. Minor revisions as suggested below are necessary before this article be considered for publication

  • The authors provide no explanation as to why the HA∆bind generated is an H3 HA(HK68) while the two other viruses used (NAbind/HAwt, NAwt/HAwt) are H1 HAs (Cal09). Similar mutations have been described in Cal09 (Y108F, Reference 1) and moreover the Y98F mutation is well described in other Group 1 HAs (References 2, 3). The rationale for the generating and H3N1 NAbind/HA∆bind should be addressed explicitly or better yet a modified H1N1 viruses should be generated and assessed.
  • Along similar lines, to demonstrate that HA sialic acid binding is indeed ablated in the NAbind/HA∆bind, a simple HAI should be performed.
  • It would be helpful to include a summary table that describes the epitopes for the monoclonal antibodies utilized, as well as their relative NA inhibition in ELLA and single-cycle infection assays (i.e. + weakly neutralizing, +++ potent neutralization)
  • Line 51 typo – missing words (s) “N1 NA to bind receptor”

References:

  • Leon PE, He W, Mullarkey CE, et al. Optimal activation of Fc-mediated effector functions by influenza virus hemagglutinin antibodies requires two points of contact. Proc Natl Acad Sci U S A. 2016;113(40):E5944‐E5951. doi:10.1073/pnas.1613225113
  • Whittle JR, Wheatley AK, Wu L, et al. Flow cytometry reveals that H5N1 vaccination elicits cross-reactive stem-directed antibodies from multiple Ig heavy-chain lineages. J Virol. 2014;88(8):4047‐4057. doi:10.1128/JVI.03422-13
  • Martín J, Wharton SA, Lin YP, et al. Studies of the binding properties of influenza hemagglutinin receptor-site mutants. Virology. 1998;241(1):101‐111. doi:10.1006/viro.1997.8958

Author Response

Responses to Reviewer 2 are written in italic:

As compared to hemagglutinin, neuraminidase and antibodies recognizing neuraminidase (NA) have received decidedly less attention. Yet, it is increasingly clear that a better understanding of NA-based immunity has important implications for the development of novel therapeutics. Innovative tools and techniques to study anti-NA antibodies are needed to help fill our gaps in knowledge. In the study described here, Gentles and colleagues describe a modified influenza A virus that is dependent on NA for entry. The experiments that follow are straightforward. Using a single-cycle infection assays, the authors demonstrate that a panel of anti-NA antibodies can neutralize this HA-binding deficient virus. The authors then go on to demonstrate that this modified IAV can be used to select antibody escape mutant for NA. The latter use offers perhaps the greatest utility going forward, given that enzyme-linked lectin assays (ELLA) can be used to identify antibodies that inhibit NA enzymatic activity. Minor revisions as suggested below are necessary before this article be considered for publication.

The authors provide no explanation as to why the HA∆bind generated is an H3 HA(HK68) while the two other viruses used (NAbind/HAwt, NAwt/HAwt) are H1 HAs (Cal09). Similar mutations have been described in Cal09 (Y108F, Reference 1) and moreover the Y98F mutation is well described in other Group 1 HAs (References 2, 3). The rationale for the generating and H3N1 NAbind/HA∆bind should be addressed explicitly or better yet a modified H1N1 viruses should be generated and assessed.

The H3 HA∆bind has been rigorously tested in our lab showing that this combination of mutations completely prevents HA binding while still allowing for efficient viral growth in the presence of a receptor-binding NA. To our knowledge, this is the only HA that has been characterized in the literature as completely binding deficient. Although Y98F and other single mutations to HA greatly reduce receptor affinity, they retain enough binding activity to enable the virus to still grow in cell culture (although not mice); see for instance https://jvi.asm.org/content/82/10/5079. In our original work generating the H3 HA∆bind, we found it was necessary to introduce a large number of mutations (three receptor-binding point mutations, deletion of the 220 loop, and extra glycosylation) to make HA so completely binding deficient that it could not support any viral growth in the absence of a binding NA. We agree that it should be possible to generate a similar H1 HA, but since it requires engineering the HA so extensively we chose to just use the H3 HA. In particular, since all antibodies tested here are directed to the NA rather than HA removing any dependency on HA antigenicity, we felt it did not matter for the purpose of these experiments if the HA was H1 or H3.

We have added text in the Results explaining these facts.

Along similar lines, to demonstrate that HA sialic acid binding is indeed ablated in the NAbind/HA∆bind, a simple HAI should be performed.

The ablation of HA binding in the NAbind/HA∆bind virus identical to the one used here, and the lack of receptor-binding of this HA has been rigorously demonstrated by Hooper et. al 2015. That point is now cited clearly.

It would be helpful to include a summary table that describes the epitopes for the monoclonal antibodies utilized, as well as their relative NA inhibition in ELLA and single-cycle infection assays (i.e. + weakly neutralizing, +++ potent neutralization)

Added suggested table and refer to table in text.

Line 51 typo–missing words (s) “N1 NA to bind receptor”

Fixed wording for readability.
